# Recovery of Graphite and Cathode Active Materials from Spent Lithium-Ion Batteries by Applying Two Pretreatment Methods and Flotation Combined with a Rapid Analysis Technique

Hao Qiu [1],*, Christoph Peschel [2], Martin Winter [2,3], Sascha Nowak [2], Johanna Köthe [1] and Daniel Goldmann [1]

1   Institute of Mineral and Waste Processing, Waste Disposal and Geomechanics (IFAD),
    Clausthal University of Technology, Walther-Nernst-Str. 9, 38678 Clausthal-Zellerfeld, Germany;
    johanna.koethe@tu-clausthal.de (J.K.); daniel.goldmann@tu-clausthal.de (D.G.)
2   MEET Battery Research Center, University of Münster, Corrensstr. 46, 48149 Münster, Germany;
    christoph.peschel@uni-muenster.de (C.P.); martin.winter@uni-muenster.de (M.W.);
    sascha.nowak@uni-muenster.de (S.N.)
3   Helmholtz-Institute Münster, IEK-12, Forschungszentrum Jülich, Corrensstraße 46, 48149 Münster, Germany
*   Correspondence: hao.qiu@tu-clausthal.de

**Abstract:** This work investigates the comprehensive recycling of graphite and cathode active materials ($LiNi_{0.6}Mn_{0.2}Co_{0.2}O_2$, abbreviated as NMC) from spent lithium-ion batteries via pretreatment and flotation. Specific analytical methods (SPME-GC-MS and Py-GC-MS) were utilized to identify and trace the relevant influencing factors. Two different pretreatment methods, which are Fenton oxidation and roasting, were investigated with respect to their influence on the flotation effectiveness. As a result, for NMC cathode active materials, a recovery of 90% and a maximum grade of 83% were obtained by the optimized roasting and flotation. Meanwhile, a graphite grade of 77% in the froth product was achieved, with a graphite recovery of 75%. By using SPME-GC-MS and Py-GC-MS analyses, it could be shown that, in an optimized process, an effective destruction/removal of the electrolyte and binder residues can be reached. The applied analytical tools could be integrated into the workflow, which enabled process control in terms of the pretreatment sufficiency and achievable separation in the subsequent flotation.

**Keywords:** flotation; Fenton oxidation; thermal pretreatment; lithium-ion battery; recycling; graphite; SPME; GC-MS

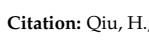



## 1. Introduction

With the expanding market of electric vehicles, the production of Lithium-ion batteries (LIBs) is booming, which has resulted in an increasing quantity of end-of-life (EoL) LIBs annually [1,2]. Furthermore, since spent LIBs contain considerable high-economic-value metals and hazardous materials, the comprehensive recycling of EoL LIBs is a meaningful endeavor from a circular economy perspective [3–7].

Currently, the recycling technologies of EoL LIBs mainly include pyrometallurgical processing [8–12], pyrolysis [13–16], mechanical treatment [17,18], and hydrometallurgical processing [19,20]. These paths do not stand alone in a complete recycling process but are combined. Among several typical recycling processes, the Accurec process [19] includes pyrolysis-mechanical treatment and hydrometallurgical processing, and the Duesenfeld process [19] combines mechanical process treatment and hydrometallurgy. The main products of the existing recycling technologies and processes are valuable metals, such as nickel, cobalt, copper, and aluminum [4]. Graphite has not been of significant interest in recycling yet, although it is a critical raw material that is defined by the European Commission [2,21]. From an economic point of view, the price of battery-grade graphite has reached USD 8–13/kg, and the graphite content in LIBs is 13.8–22% by weight [2,4]. In addition, the early separation of graphite from the cathode active material can improve the

efficiency of the subsequent hydrometallurgical process that is applied to recover valuable metals [21]. In current processes, the most economically relevant constituents that are used in cathode and anode material, plus the conducting salt, are separated from the other components in the black mass fraction.

Black mass (BM) refers to the mixture that is detached from the current collector foils during mechanical treatment. It mainly consists of the cathode active material, such as NMC (LiNi$_x$Mn$_y$Co$_{1-x-y}$O$_2$) particles, and the anode active material graphite or graphite/silicon composite from spent LIBs. Trying to recover the valuable metals and, in addition, a cleaned graphite fraction flotation, before leaching seems to be a favorable approach, separating the cathode from the anode material for the subsequent individually adapted chemical separation steps. Froth flotation is a conventional technique that is used for enriching and separating different minerals on the basis of the surface hydrophobic differences [22]. Because natural graphite and pure NMC particles have significant wettability differences [2], many researchers currently utilize flotation methods in their studies for the separation of NMC from graphite. However, Zhang et al. found that the surface of the electrode active material particles was covered with organic substances that originated from the used electrolyte in the battery system, which made the wettability difference between the cathode active material and the graphite particle surfaces very similar, which thus reduced the separation efficiency of the flotation [23,24]. On the basis of these findings, chemical or thermal pretreatment steps on the BM should be applied to overcome this problem.

For chemical pretreatment, Fenton oxidation is an option. Fenton oxidation belongs to one of the advanced oxidation processes (AOPs), which generate hydroxyl radicals by adding Fe$^{2+}$/H$_2$O$_2$ [25]. The strong oxidation ability of hydroxyl radicals can degrade large molecules of organic substances into small molecules or less harmful compounds, which are often used in wastewater treatment [25]. He et al. report the use of Fenton oxidation to remove organic substances from the outer layer of cathode and anode active material, followed by flotation, to obtain LiCoO$_2$ with a Co grade of 39.91% and a 98.99% recovery [5].

An alternative could be a thermal treatment step that is dedicated to destroying the organic coating substances. Yang et al. utilized roasting to pretreat the BM [21]. The graphite recovery from the froth product of the following flotation can exceed 90% at a roasting temperature of 350 °C–450 °C and a roasting time of 30 min, but its grade is around 50%. This also indicates that a large portion of the cathode material goes into the froth product, along with the graphite, which may be caused by the incomplete removal of organic residues from the particle surface. Wang et al. report that they obtained LiCoO$_2$ with a grade of 40.12% and a recovery of 97.66% by using roasting-assisted flotation [26]. Zhang et al. report the use of pyrolysis to remove organic residues from the surface of the particles when the pyrolysis temperature reached 550 °C [6]. This was followed by flotation to obtain LiCoO$_2$ with a grade of 94.72% and an 83.75% recovery. Vanderbruggen et al. report attrition on pyrolyzed BM as a pretreatment to achieve a 70–85% LMO recovery and an 86% graphite recovery via flotation [27]. A thermal treatment that goes in the opposite direction would be a cryogenic pretreatment. Liu et al. used cryogenic grinding to remove organic residues from the particle surface, and the grade of the pulp product LiCoO$_2$ of the flotation reached 91.75%, with a recovery of 89.83% [3]. The results of recent studies are summarized in Table 1.

To evaluate the effectiveness of a pretreatment preceding the flotation, different analytical tools could be applied. Analysis methods, such as thermal gravimetric analysis (TGA) or gas chromatograph—mass spectrometry (GC-MS) with solid phase microextraction (SPME), or GC-MS with a pyrolysis sample introduction (Py-GC-MS), can identify and determine the loss of this organic matter in the dependency of the applied treatment [28–30].

**Table 1.** Summarized flotation results of recent studies.

| Method and Reference | Graphite Grade (Froth Product), % | Graphite Recovery (Froth Product), % | LiCoO₂ Grade (Pulp Product), % | LiCoO₂ Recovery (Pulp Product), % |
|---|---|---|---|---|
| Fenton-oxidation-assisted flotation by He et al. [5] | - | - | 66 (Co content: 39.91) | 98.99 |
| Roasting-assisted flotation by Yang et al. [21] | around 50 | >90 | - | around 50 |
| Roasting-assisted flotation by Wang et al. [26] | - | - | 67 (Co content: 40.12) | 97.66 |
| Pyrolysis-assisted flotation by Zhang et al. [6] | - | - | 94.72 | 83.75 |
| Attrition-assisted flotation by Vanderbruggen et al. [27] | 63.1–74.2 | 86 | - | (LMO) 70–85 |
| Cryogenic-grinding-assisted flotation by Liu et al. [3] | - | - | 91.75 | 89.83 |

Most of these current studies on preflotation treatment focus on NMC as the target, and not many studies have been conducted that consider the grade and recovery of both graphite and NMC from flotation processes. In this work, comprehensive recycling was considered (i.e., the recovery and grade of the two target products—graphite and NMC—were taken into account). Firstly, statistical experiments were designed to investigate the effect of no pretreatment on the flotation effect of the BM. Secondly, two existing pretreatment methods, Fenton oxidation and roasting, were investigated with respect to their influence on the flotation effectiveness. In the Fenton oxidation pretreatment experiments, the effect of the pH and the concentration of the Fenton's reagent on the flotation products was investigated. In the roasting pretreatment experiments, the effect of the roasting temperature and the roasting time was investigated. After that, flotation optimization experiments were performed to investigate the effects of different flotation conditions, such as the reagent dosage, on the flotation effectiveness.

More importantly, the current methods for analyzing the organic residues on BM surfaces, such as X-ray photoelectron spectroscopy (XPS), which is slow and costly, are not efficient enough or are lacking the needed limits of detection. Another common indirect method is to perform flotation experiments first, and to then combine the elemental analysis results of the flotation products to assess whether the organic residues are removed, which is also not reliable.

Therefore, SPME-GC-MS and Py-GC-MS analyses were newly integrated into the workflow to analyze the effectiveness of different BM treatments. These techniques enabled the determination of the molecular structure of the residues and, furthermore, the repeated measurement of the BM after treatment was utilized to control the sufficient removal of previously determined species by the applied conditions. Finally, the correlations between the identified organic residues, preceding the pretreatment and the achievable flotation efficiency, were concluded.

This study includes both graphite and NMC as products for recovery and it integrated a rapid analytical method into the workflow, which aimed to further improve the overall recovery rate and the efficiency of the recovery process, which has important implications for the refinement of the existing mechanical-hydrometallurgical route to the recovery of EoL LIBs. It is also an essential component in the construction of the circular economy.

## 2. Materials and Methods

### 2.1. Experimental Materials

NMC-622 EoL LIBs (modules) from an industrial partner served as the test material. The material was shredded under vacuum conditions, which removed low-boiling components from the electrolyte.

The experimental material was firstly subjected to sieve analysis. After splitting the samples, 150 g of the subsamples was screened with sieves of 2, 1, 0.5, 0.25, 0.1, and 0.063 mm. A piece of ultrasonic-assisted equipment HAVER UFA (HAVER & BOECKER, Oelde, Germany) was added to the 0.1 mm and 0.063 mm sieves to ensure the effective sieving of the fine-grained fractions. The sieving time was 30 min, and if the weight of the undersize product after manual re-sieving (1 min) did not exceed 1% of the oversize mass, the sieving was finished. The individual fractions that were obtained after sieving were analyzed separately by ICP-OES (ICP-OES 5100, Agilent, Waldbronn, Germany).

For the pretreatment and flotation experiments, about 5 kg of shredded materials was sieved through a 0.25 mm sieve, and the undersize of a 0.25 mm BM fraction was selected.

### 2.2. Methods

#### 2.2.1. Fenton Pretreatment

Firstly, two different concentrations of Fenton's reagents, F1 and F2, were prepared. A 0.1 M $FeSO_4$ (Riedel-de-Häen, Seelze, Germany) solution and a 0.1 M $H_2O_2$ (30%, CARL ROTH, Karlsruhe, Germany) solution were prepared and mixed at a volume ratio of 120:1 in order to obtain F1. For F2, a 0.5 M solution of FeSO4 and a 0.5 M solution of $H_2O_2$ were prepared and mixed in a volume ratio of 120:1. Afterward, F1 and F2 were added to the samples at a 40 g/L solid-to-liquid ratio (S/L ratio) and were stirred at 250 rpm and at the desired pH. The stirring time for each sample was 30 min. The target pH (1–7) needed to be kept constant during the experiment.

The pretreated samples were subsequently filtered, dried, and subjected to SPME-GC-MS analysis, Py-GC-MS analysis, and flotation experiments.

#### 2.2.2. Roasting Pretreatment

Preliminary experiments were first performed with respect to the roasting temperature. The fine fraction undersize (0.25 mm) was placed in two ceramic crucibles and roasted in a muffle furnace (Nabertherm, L9/11/SKM, Lilienthal, Germany) The crucibles were placed in the furnace at ambient temperature and were then heated to 200 and 450 °C, respectively. After the temperature had been reached, the material remained in the furnace for a further 30 min in each case. After roasting, the samples were then measured with Py-GC-MS and SPME-GC-MS.

Subsequently, the holding time was further investigated on the basis of a heating temperature of 450 °C. Three ceramic crucibles were filled with 15 g of sample each, and the muffle furnace was heated to 450 °C. When the temperature reached 450 °C, the samples were placed and held for 30, 45, and 60 min, respectively. Finally, flotation was performed with these treated samples.

#### 2.2.3. Flotation Experiment

Flotation experiments (1-stage) were conducted in a 125 mL Denver-type flotation machine in an IFAD construction (Figure 1). The collector that was used was ShellSol$^{TM}$ D100 (Shell, London, UK), with $C_{13}$–$C_{15}$ paraffins and naphthenes as the main components [31]. An emulsifier (Emulsogen EL, Hoechst, Frankfurt, Germany) was added to the collector in a mass ratio of 1:10 and was diluted to a 1 wt% emulsion for use. MIBC (Sigma-Aldrich, St. Louis, MO, USA) was employed as the frother, the inhibitor was oxalic acid (CARL ROTH, Karlsruhe, Germany), and the dispersant was sodium hexametaphosphate (Riedel-de-Häen, Seelze, Germany). MIBC, oxalic acid, and sodium hexametaphosphate were diluted to a 1 wt% solution for use. The stirring speed was 2000 rpm, and the airflow rate was 40 L/h. The conditioning time was one minute for each reagent, and the flotation scraping time was 5 min.

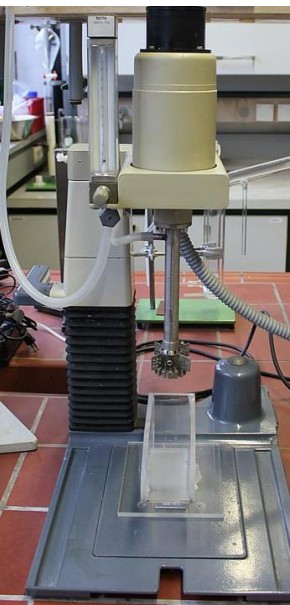

**Figure 1.** IFAD-construction Denver-type flotation machine.

Four-factor (variable) and two-level full factorial experiments were designed in the flotation experiments performed directly without pretreatment. At the same time, five center points were inserted as replicates to estimate the experimental error and to detect the curvature. The four variables are: Factor A—the collector dosage; Factor B—the frother dosage; Factor C—the inhibitor dosage; and Factor D—the dispersant dosage (Table 2).

**Table 2.** Variables for the full factorial design. "1" corresponds to a high factor level; "−1"refers to a low factor level; "0" refers to a center point.

| Variable | Coded Levels | Variable Range |
|---|---|---|
| Factor A: Collector dosage | 1 | 550 g/t |
| | 0 | 350 g/t |
| | −1 | 150 g/t |
| Factor B: Frother dosage | 1 | 550 g/t |
| | 0 | 350 g/t |
| | −1 | 150 g/t |
| Factor C: Inhibitor dosage | 1 | 300 g/t |
| | 0 | 150 g/t |
| | −1 | 0 g/t |
| Factor D: Dispersant dosage | 1 | 300 g/t |
| | 0 | 150 g/t |
| | −1 | 0 g/t |

The purpose of the experiments was to determine whether these four factors and their interaction effects influenced the flotation efficiency of the BM without any pretreatment. Pareto charts were used to determine the significance of the main and interaction effects of the factors, and to exclude insignificant factors [32,33]. The experimental design and statistical analysis were performed by using the Design-Expert® Software (version 10.0.8, Stat-Ease, Inc., Minneapolis, MN, USA).

Only the collector and the frother were used in the Fenton-pretreatment-assisted flotation experiments. The slurry concentration was 7.4%, the collector dosage was 300 g/t, and the frother dosage was 150 g/t. The conditioning time was 1 min, and the flotation scraping time was 5 min.

After that, the effect of the roasting time on the flotation products was studied. The roasting pretreatment combined flotation experiments used only the collector and frother. The slurry concentration was 7.4%, the collector dosage was 150 g/t, the frother dosage was 150 g/t, the conditioning time was 1 min, and the scraping time was 5 min.

In the subsequent single-factor optimization experiments for the roasted BM, the factor collector dosage (50, 100, 150, and 300 g/t), the frother dosage (50, 100, 150, and 300 g/t), the slurry concentration (3.8, 7.4, 10.7, and 13.8%), and the airflow rate (20, 40, and 60 L/h) were investigated in order to optimize the flotation efficiency.

### 2.2.4. Analytical Methods

The metal element analysis of the BM was performed by ICP-OES (ICP-OES 5100, Agilent, Waldbronn, Germany), while the graphite grade was determined by measuring the total carbon content (elemental analyzer EA4000, Analytik Jena, Jena, Germany). The theoretical nickel content in NMC-622 ($LiNi_{0.6}Mn_{0.2}Co_{0.2}O_2$) is 36.6%, and the analysis of the nickel content of the reference material, NMC-622 (BASF, Ludwigshafen, Germany), showed that the nickel content was 37.4%. Therefore, the nickel content was used as an indicator to determine the NMC grade in the pulp products, and the average value of 37% was taken as the highest theoretical purity that was achievable for the nickel in the flotation product.

Gas-chromatography-based analyses were executed on a Shimadzu (Kyoto, Japan) GCMS-QP2010 Ultra that was assembled with a nonpolar Supelco SLB®-5 ms (30 m × 0.25 mm. 0.25 μm; Sigma Aldrich, St. Louis, MO, USA) column. SPME (acrylate fiber, CTC analytics, Zwingen, Switzerland) and pyrolysis (PY-3030D pyrolyzer, Frontiers Laboratories, Tokyo, Japan) were utilized for the analyte injection from the solid recycling material. Pyrolysis was performed with three or four successive steps at 200, 300, (450) and 515 °C. Detailed chromatographic parameters were utilized, according to previous studies [28,29].

## 3. Results and Discussions

### 3.1. Particle Size and Elemental Distribution

After sieving, the particle size distribution of the BM was obtained, and it is shown in Figure 2. The sieve analysis showed that the main mass fractions were distributed among the sieve classes < 0.063 mm, >2 mm, and 1–2 mm.

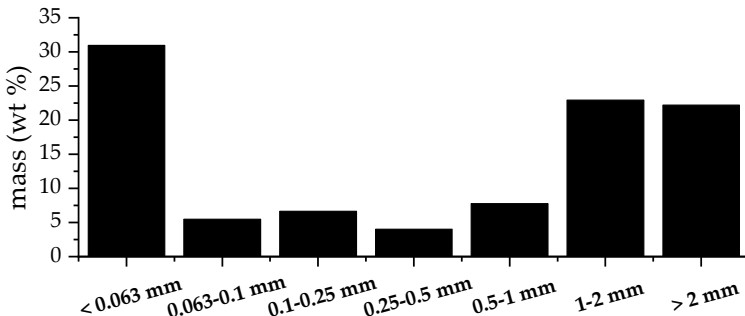

**Figure 2.** Particle size distribution.

The main metal content in all the fractions is shown in Figure 3. Three size fractions (0.1–0.25 mm, 0.063–0.1 mm, and <0.063 mm) had lower total metal contents than the other fractions. This was probably caused by the increased carbon content in these fine-sized fractions. In addition, the distribution of the aluminum and copper was not homogeneous in the different particle size fractions. In the 0.25–0.5 mm fraction, the Al content was 11%, and the Cu content was 23%, while, in the 0.1–0.25 mm fraction, the Al content decreased to 3%, and the Cu content dropped to 4%. In these three fine-sized fractions (0.1–0.25 mm, 0.063–0.1 mm, and <0.063 mm), the Al and Cu contents were significantly lower than in the coarser fractions.

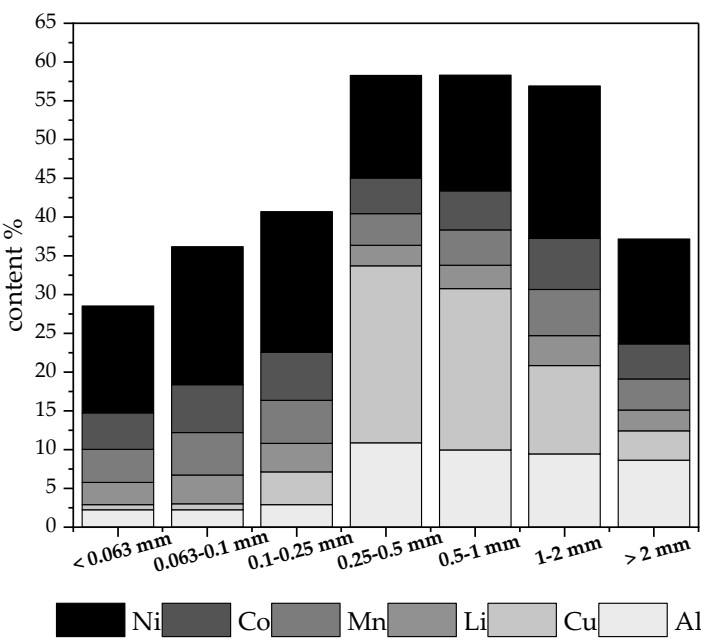

**Figure 3.** Distribution of main metal contents in different size fractions.

### 3.2. Flotation of Untreated Black Mass

To determine the organic electrolyte and binder residues of the starting material, SPME-GC-MS and Pyrolysis-GC-MS were utilized, respectively. (Figures 4 and 5)

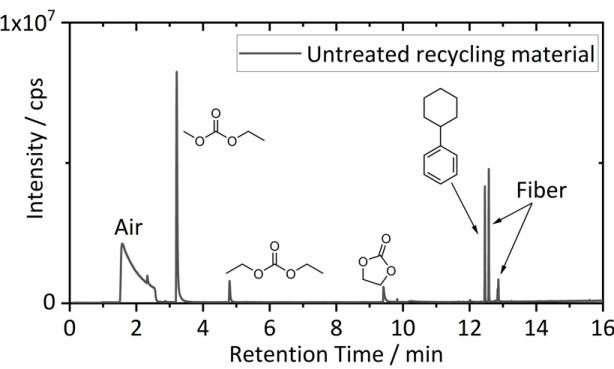

**Figure 4.** SPME-GC-MS chromatogram after preconcentration for 10 s from the headspace above the untreated recycling material.

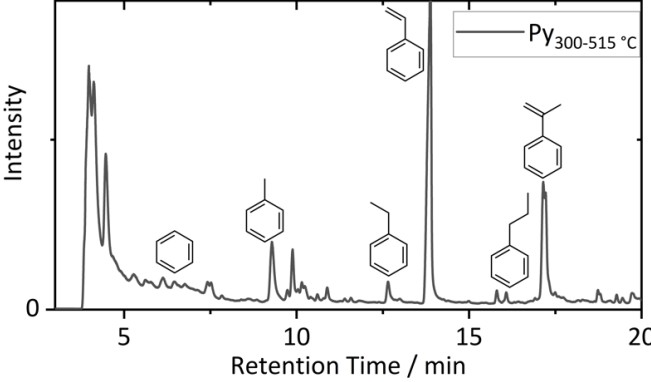

**Figure 5.** Py-GC-MS pyrogram (515 °C pyrolysis temperature) of the untreated shredded material. SBR-originating compounds are marked. The shown "fingerprint" was used for the targeted identification of the SBR in the treated material.

The qualitative SPME-GC-MS measurement of the untreated recycling material showed ethyl methyl carbonate (EMC), diethyl carbonate (DEC), ethylene carbonate (EC) and cyclohexylbenzene as the main detectable electrolyte residues after a short preconcentration duration (10 s). Furthermore, air- and fiber-caused system peaks were found, which were not related to the sample constitution.

The Py-GC-MS investigations resulted in the characteristic and literature-described binder fingerprints, which are exemplarily shown for styrene butadiene rubber (SBR) [28,34]. SBR is used as a binder for graphitic anodes, and not only because of its high decomposition temperatures, but also because of its characteristic Py-GC-MS fingerprint. It was chosen to further visualize the pretreatment efficiency of the BM.

For this material, a set of factorial experiments was designed to determine the effect of four factors (A: the collector dosage; B: the frother dosage; C: the inhibitor dosage; and D: the dispersant dosage) on the untreated material. In Table 3, four factors and four responses are presented for the full factorial experiments (i.e., R1: the C content in the froth product; R2: the graphite recovery in the froth product; R3: the nickel content in the pulp product; and R4: the recovery of the NMC in the pulp product).

**Table 3.** Full factorial design matrix and responses. "1" corresponds to a high factor level; "−1" refers to a low factor level; and "0" refers to a center point.

| Experimental Trial Number | A | B | C | D | R1, % | R2, % | R3, % | R4, % |
|---|---|---|---|---|---|---|---|---|
| 1 | 1 | 1 | −1 | 1 | 52 | 95 | 23 | 17 |
| 2 | −1 | −1 | −1 | −1 | 51 | 94 | 25 | 23 |
| 3 | −1 | 1 | −1 | 1 | 53 | 84 | 22 | 37 |
| 4 | 0 | 0 | 0 | 0 | 53 | 88 | 24 | 33 |
| 5 | 1 | −1 | −1 | 1 | 52 | 89 | 24 | 31 |
| 6 | 1 | 1 | −1 | −1 | 52 | 90 | 23 | 27 |
| 7 | 0 | 0 | 0 | 0 | 52 | 85 | 21 | 30 |
| 8 | 0 | 0 | 0 | 0 | 52 | 87 | 22 | 29 |
| 9 | 1 | −1 | 1 | −1 | 55 | 84 | 23 | 42 |
| 10 | −1 | −1 | 1 | −1 | 55 | 88 | 23 | 33 |
| 11 | 0 | 0 | 0 | 0 | 50 | 95 | 24 | 18 |
| 12 | 1 | 1 | 1 | 1 | 52 | 88 | 23 | 29 |
| 13 | −1 | 1 | 1 | −1 | 56 | 79 | 23 | 46 |
| 14 | −1 | −1 | −1 | 1 | 53 | 87 | 25 | 36 |
| 15 | 0 | 0 | 0 | 0 | 53 | 90 | 26 | 36 |
| 16 | 1 | −1 | 1 | 1 | 51 | 92 | 25 | 29 |
| 17 | −1 | 1 | 1 | 1 | 57 | 86 | 25 | 41 |
| 18 | −1 | −1 | 1 | 1 | 55 | 91 | 24 | 29 |
| 19 | −1 | 1 | −1 | −1 | 55 | 88 | 24 | 39 |
| 20 | 1 | −1 | −1 | −1 | 55 | 86 | 23 | 36 |
| 21 | 1 | 1 | 1 | −1 | 57 | 86 | 24 | 42 |

From Table 3, it is clear that, although the recovery of graphite in the froth product (R2: 79–95%) was high, the grade of the graphite (R1: 50–57%) was low, which was due to the entrapment of a large amount of cathode active material, which also led to the low recovery of NMC (R4: 17–46%) in the pulp product. Furthermore, the enrichment ratio of graphite in the froth product in this whole set of experiments was only 1.1–1.2. The NMC grades in the pulp product can reach approximately 58–69% (R3: 21–26%).

A Pareto chart can be used to determine the essential effects and the magnitude [35]. The length of each bar is proportional to the absolute value of its standardized effects, and the bars are ranked as the size of the effect [35–38]. The horizontal line refers to a test at $p = 0.05$ and a t-value = 2.086. A factor is statistically significant when the absolute value of the standardized effect of the factor is above the line [36]. As is shown in Figure 6A, for R1, only the absolute value of the effect of Factor C was slightly higher than the t-value limit. For Responses 2 and 3 (Figure 6B,C), no significant effects were found. For Response 4

(Figure 6D), only the absolute value of the interaction effect of Factor AB was slightly higher than the t-value limit. The omission of some important factors may cause the insignificant effect of nearly all the factors. Considering the results of the SPME-GC-MS and Py-GC-MS, the important omission factor can be the removal level of the electrolyte and the binder.

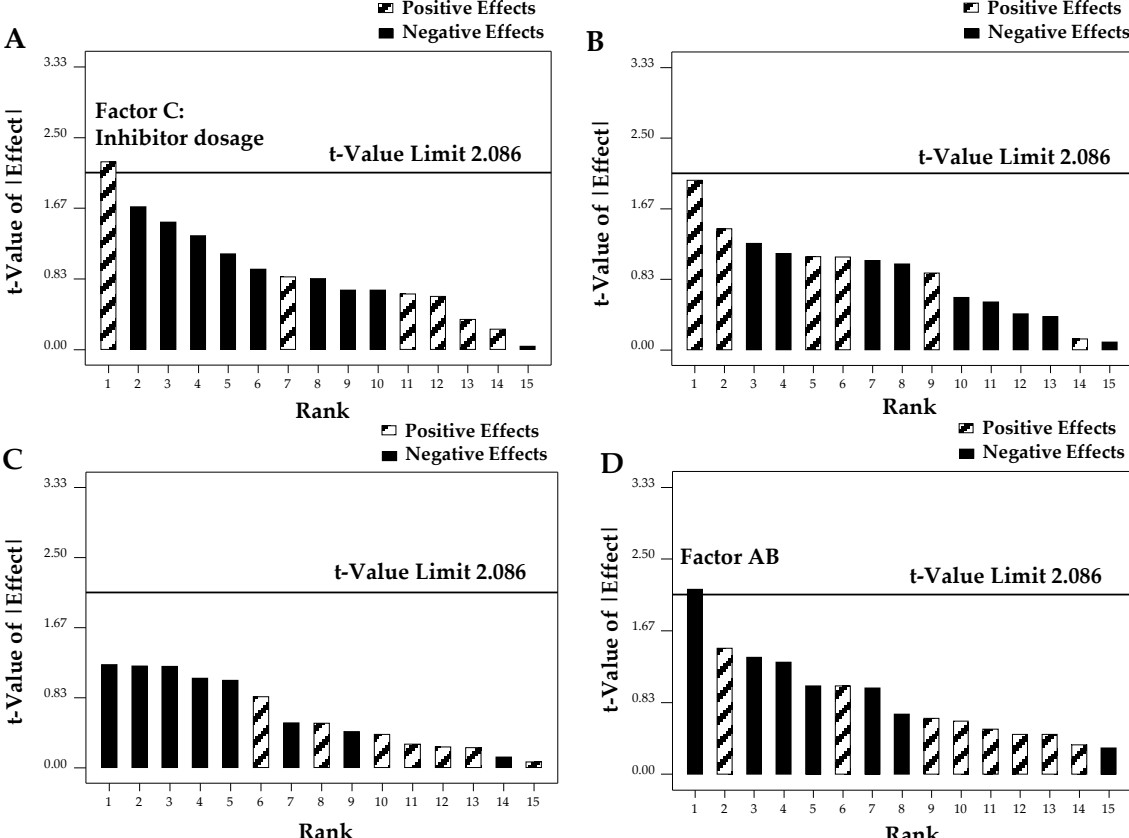

**Figure 6.** Pareto chart of the absolute values of the standardized effects of the factors for the 4 responses: (**A**) R1: the C content in the froth product; (**B**) R2: the graphite recovery in the froth product; (**C**) R3: the nickel content in the pulp product; and (**D**) R4: the recovery of NMC in the pulp product.

The above also indicate that the separation efficiency of the graphite and the NMC by flotation from the untreated BM was unsatisfactory when both the grade and the recovery of the graphite and the NMC were included in the evaluation system of the flotation effectiveness. If the flotation was performed directly for the untreated BM, although a high graphite recovery was achieved, the graphite grade was not high (50–57%). More importantly, the recovery of the NMC was low (17–46%).

### 3.3. Effects of Fenton Pretreatment Parameters on Flotation Efficiency

The pH is considered to be an essential factor that affects the Fenton reaction [25,39]. Recent studies have shown that a pH < 2.5 leads to the formation of $[Fe(H_2O)_6]^{2+}$, and it thus generates a lower amount of hydroxyl radicals [39], while, at a basic pH, the Fenton reaction is interrupted because of the precipitation of the iron ions [40]. Using density functional theory calculations (DFT), Lu et al. report that, at a pH of 2.5–3.5, the major oxidant is $[(H_2O)_5Fe^{IV}O]^{2+}$, which is highly reactive as the hydroxyl radical. As the pH increases, the less reactive $[(H_2O)_4Fe^{IV}O(OH)]^+$ and $[(H_2O)_3Fe^{IV}O(OH)_2]$ then become the dominant oxidants [41].

Therefore, the optimal operating pH is usually considered to be around 3 [39,42–44]. Lin et al. report that the optimum pH is 3 when treating simulated textile wastewater [42]. Guedes et al. observed the maximum total organic carbon removal under a pH of 3.2

by treating cork-cooking wastewater [43]. However, the introduction of chelating agents allows the Fenton reaction to proceed at higher pH values [40,45].

In this section, the effects of the pH (1–7) and the concentration of the Fenton's reagent on the flotation efficiency is investigated.

At the beginning, the BM was pretreated with 0.1 M of Fenton reagent at different pH values. Afterward, the flotation experiments were carried out. The flotation results (Figure 7A,B) show that the separation of the NMC and the graphite was not significant. For example, the grade of the graphite was about 52%, and the graphite recovery was 81% if the pH was controlled at 3 and 4 during the pretreatment. On the other hand, the recovery of the NMC in the pulp product was around 30%, with an NMC grade of 51% (Ni-content: 19%). The flotation results of almost all of the samples from pH 1 to pH 7 were close to those of the untreated BM (Table 3) (i.e., although the recovery of the graphite in the froth product was high, the grade of the graphite was low because of a large amount of entrapped NMC particles).

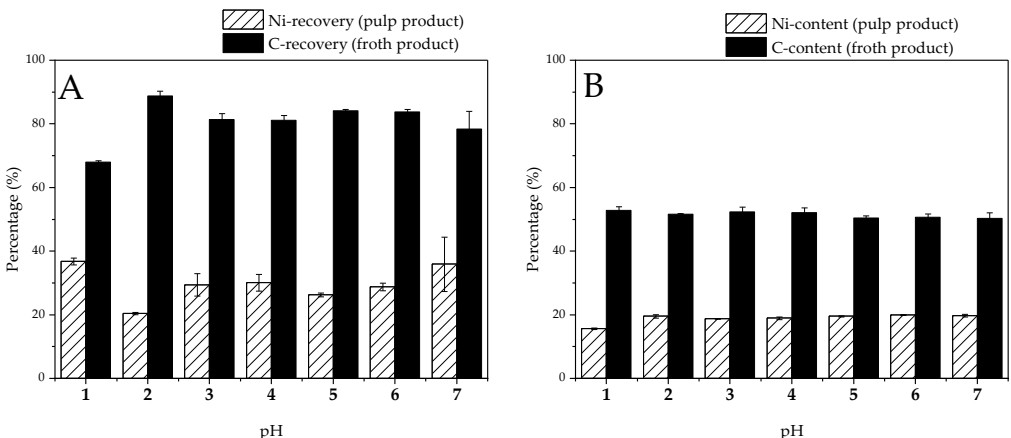

**Figure 7.** (**A**) Recoveries and (**B**) contents of flotation products obtained from the BM pretreated with F1 solution (0.1 M) at different pH values.

By contrast, the recovery of the NMC in the pulp product was low. This may be due to the low concentration of the Fenton's reagent and the high S/L ratio, which affected the removal of the organic residues. In addition, at this concentration, the pH environment during the Fenton pretreatment did not significantly affect the flotation efficiency.

The flotation experiments were also conducted after the BM was pretreated with 0.5 M of Fenton's reagent at different pH values (Figure 8A,B). The results show that, when the pretreatment pH was 3, the C content in the froth product was 62%, and the graphite recovery was 46%. The recovery of the NMC in the pulp product was 76%, and the NMC grade was 46% (Ni content: 17%). When the pH was 7, the recovery of the NMC in the pulp product was 81%, and the NMC grade was 48% (Ni content: 18%), while the recovery of the graphite in the froth product was 40%, and the C content was 61%. The recovery was relatively close to that of pH 3.

Since the target products of the experiment were two products—graphite and NMC—the recovery and the grade of both products need to be included in the evaluation system of the flotation effectiveness. The experiment results illustrate an improved separation at this concentration (0.5 M). Compared with the 0.1 M Fenton's reagent pretreatment-assisted flotation results, the graphite recovery in the froth product decreased, yet the recovery of the NMC in the pulp product increased significantly. However, the recoveries were still unsatisfactory. In addition, the grades of the NMC and the graphite were not sufficient.

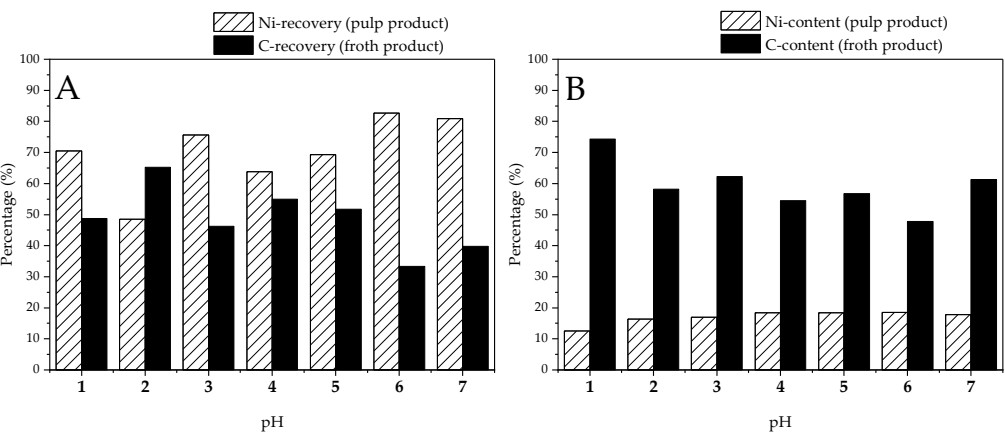

**Figure 8.** (**A**) Recoveries and (**B**) contents of flotation products obtained from the BM pretreated with F2 solution (0.5 M) at different pH values.

After all the different treatments with the Fenton reagent, no electrolyte residues were found by SPME-GC-MS. Therefore, further considerations on the treatment efficiency are based on the targeted identification of the SBR fingerprint via Py-GC-MS. After the Fenton pretreatment, three representative samples (0.5 M; pH = 1, 3, 7) were investigated, which resulted in comparable pyrograms. Exemplarily, the obtained Py-GC-MS pyrogram of pH = 3 is displayed in Figure 9, which shows insufficient binder removal, since the characteristic SBR fingerprint was still detectable.

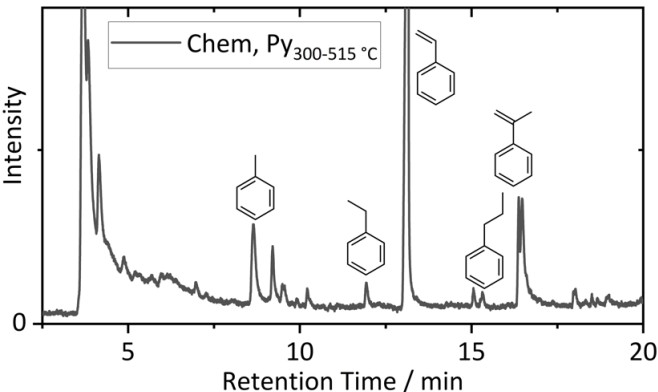

**Figure 9.** Magnified Py-GC-MS pyrogram (515 °C pyrolysis temperature) of the BM after chemical pretreatment (0.5 M; pH = 3). Characteristic SBR-originating aromatic compounds, such as styrene, are marked.

This further indicates that the removal capacity of the SBR was insufficient at both Fenton reagent concentrations. A high S/L ratio may lead to this situation. When the S/L ratio is high, the generated hydroxyl radicals are perhaps not enough to remove all of the binders. In addition, the relationship between the electrolyte removal and the flotation effect is revealed here. Compared to the results of the flotation experiments with the untreated BM, the removal of electrolytes and further residues resulted in a slight improvement in the flotation. This was reflected in the increase in the NMC recovery in the pulp product; however, at the same time, a decrease in the grade of the NMC was also observed.

### 3.4. Effects of Roasting Parameters on Flotation Efficiency

Again, no electrolyte residues were found after the thermal treatment. As expected, on the basis of the thermal stability, the Py-GC-MS investigations showed that the SBR binder was still present in the sample that was roasted at 200 °C. However, as is shown in

Figure 10, no SBR residues, or any other characteristic peaks, were found after pretreatment at 450 °C [34].

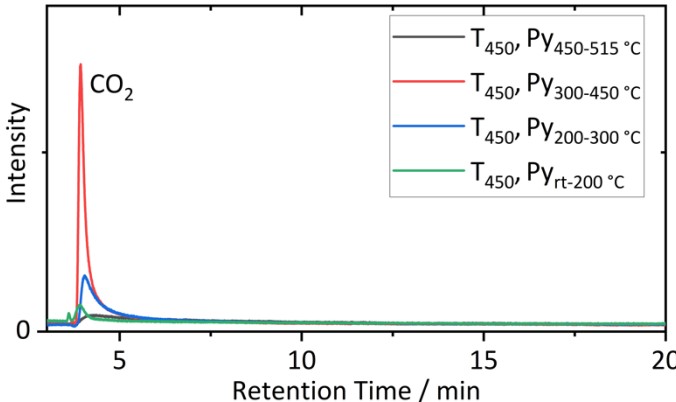

**Figure 10.** Magnified Py-GC-MS pyrograms of 4 different pyrolysis temperatures (200, 300, 450, and 515 °C) of the BM after roasting at 450 °C for 30 min.

After that, flotation experiments were performed on three samples that were obtained after different roasting times. By looking at Figure 11A,B, it is apparent that the flotation effect after roasting for 60 min was better than those with roasting times of 30 and 45 min. The graphite recovery in the froth product was 76%, and the NMC recovery in the pulp product was 90%.

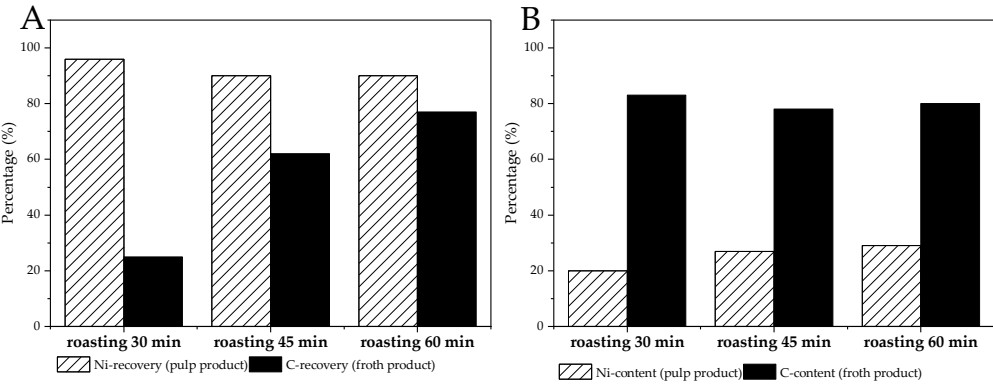

**Figure 11.** (**A**) Recoveries and (**B**) contents of flotation products obtained from the BM after different roasting times.

By comparing the calculated C content in the flotation feed, the mass loss of the graphite in the roasted product increased while the roasting time rose. For example, when the roasting time was 30 min, the C content in the roasted product was about 44%. If the roasting time was further increased to 45 min, the C content in the roasted product was about 38%, while, when the roasting time was increased to 60 min, the C content in the roasted product was about 36%.

In the end, to optimize the flotation parameters, single-factor experiments were conducted. The BM was then pretreated at 450 °C for 60 min and was soon used for the subsequent flotation experiments. The influences of the individual factors, such as the collector dosage, the frother dosage, the pulp density, and the airflow rate, were investigated, as is shown in Figures 12 and 13. The highest NMC grade (83%) (Ni-content: 31%) was obtained with an NMC recovery of 90% in the pulp product. Moreover, the C content in the froth product, of 77%, was achieved with a graphite recovery of 75% (collector: 300 g/t; frother: 150 g/t; pulp density: 7.4%; airflow rate: 40 L/h). The results show a significant increase in the selectivity after the thermal pretreatment compared to the tests on the untreated BM and the chemically pretreated BM.

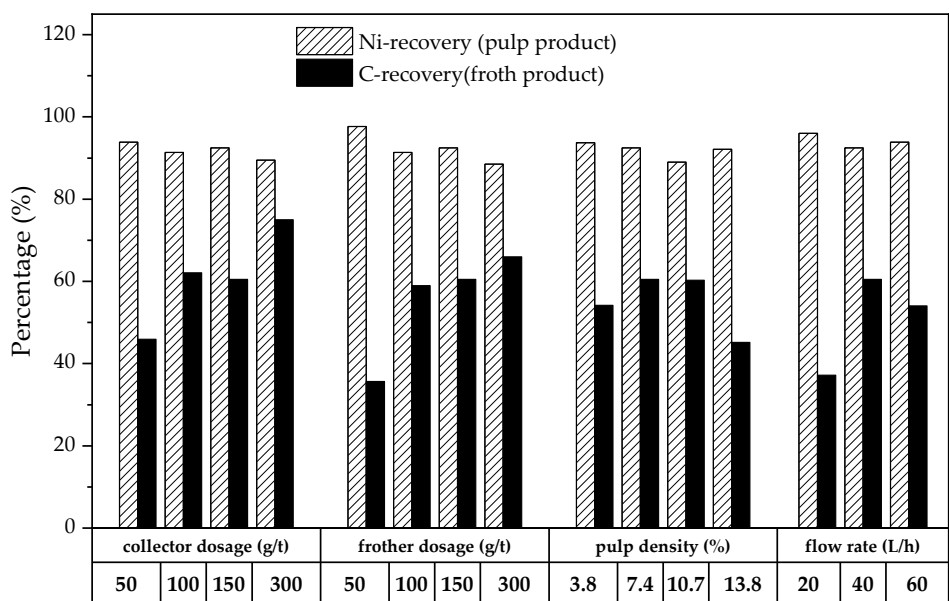

**Figure 12.** Effects of different flotation parameters on the recovery of flotation products.

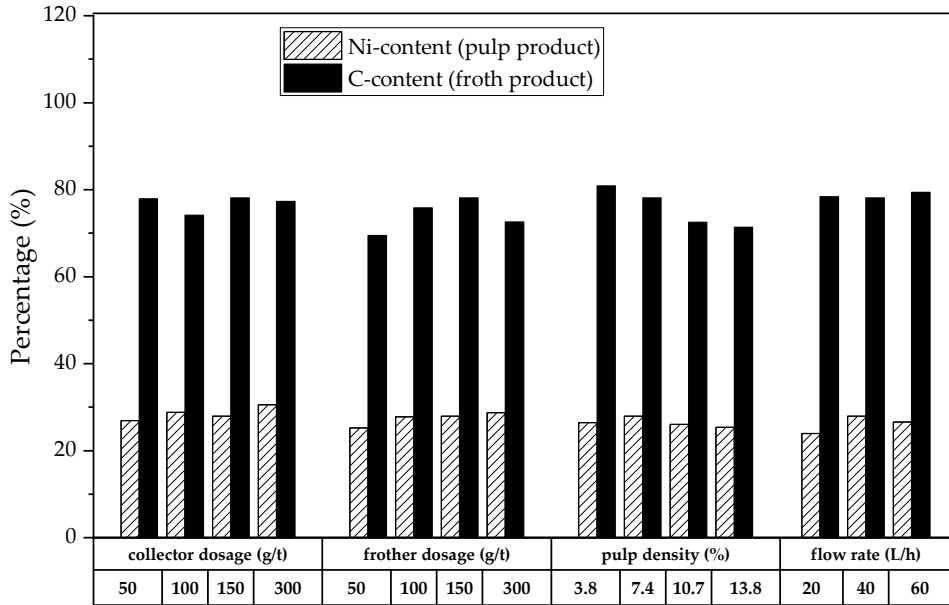

**Figure 13.** Effects of different flotation parameters on the grade of flotation products.

Combining the GC-MS analysis with the obtained flotation results, it is clear that the removal of the electrolytes and binders is the key to improving the flotation efficiency of graphite and NMC. In other words, BM treatment is inevitable. Furthermore, the mass loss of graphite and the removal of surface organic residues are related to the roasting time and to the amount of experimental material to be roasted. Even the roasting atmosphere in the furnace may affect them. Therefore, this rapid analysis method for SBR fingerprints makes it possible to determine whether the organic residues are removed from the BM and to thus predict the flotation effect.

The rapid analysis coupled with preassessment can be applied to the process control of treating the BM (Figure 14). For example, the BM that was obtained after the pyrolysis can first be analyzed to determine whether the pyrolysis time is sufficient, and if the material is ready for the subsequent flotation. Then, if further pyrolysis is needed, the pyrolysis parameters can be adjusted on the basis of the analysis results.

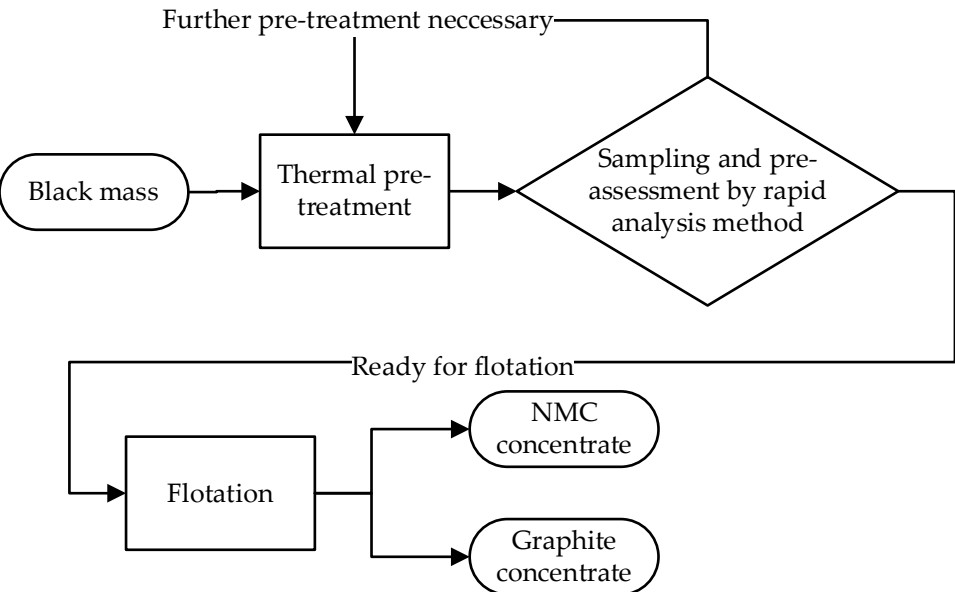

**Figure 14.** An example flow chart with the rapid analysis method integrated into the process.

## 4. Conclusions

This work investigated the comprehensive recycling of two products—graphite and NMC—via pretreatment and flotation as the central unit operations in the overall process. Two different pretreatment methods, which are Fenton oxidation and roasting, were investigated with respect to their influence on the flotation effectiveness. Furthermore, the flotation conditions for the roasting pretreated samples were optimized. In addition, GC-MS-based analyses were introduced to analyze the BM to determine whether the organic residues were removed via detection (e.g., the SBR fingerprint).

It was found that the untreated BM contained electrolyte and binder residues. Under these conditions, the graphite and NMC could not be effectively separated by flotation. As a result, although a high graphite recovery (79–95%) was achieved, the graphite grade was not satisfactory (50–57%). Moreover, the recovery of the NMC was extremely low (17–46%), and the NMC grade was 58–69%.

In the Fenton pretreatment experiment, it was found that, after pretreatment with 0.5 M of Fenton's reagent, the electrolytes residues disappeared. However, SBR binder was still found, which could be related to the high S/L ratio. Compared with the flotation results of the untreated BM, the recovery of the NMC in the pulp product rose (48–82%), and the graphite grade also increased (48–74%). In contrast, the NMC grade (34–50%) and the graphite recovery in the froth product decreased (33–65%). When both the grade and the recovery of the NMC and graphite were included in the evaluation system of the flotation effectiveness, the best flotation separation was achieved at a pH of 3. The graphite recovery was 46%, and the grade was 62%. The NMC recovery was 76%, and the grade was 46%.

In the roasting pretreatment experiments, when the roasting temperature was 450 °C and the roasting time was 30 min, no SBR binder was observed. In addition, when the roasting temperature was 450 °C, the flotation effect became better with the extension of the roasting time, but the mass loss of the graphite increased. In the subsequent flotation condition experiments, the highest NMC grade (83%) was obtained with an NMC recovery of 90% in the pulp product, and a C content in the froth product of 77% was achieved, with a graphite recovery of 75%, when both the grade and the recovery of the NMC and graphite were included in the evaluation system of the flotation effectiveness.

Overall, in this paper, roasting pretreatment combined flotation was superior to flotation with the untreated BM, and it was also superior to the Fenton-pretreatment-assisted flotation for the separation of graphite and NMC. The summarized results are

listed in Table 4. Thus, proper pretreatment is inevitable for the efficient flotation separation of the BM from spent LIBs. Herein, the introduced GC-MS-based analyses are applied for fast pretreatment process control, and it can help to presume achievable flotation separation.

**Table 4.** Summarized flotation results.

|  | Graphite Content, % | Graphite Recovery, % | NMC Content, % | NMC Recovery, % |
|---|---|---|---|---|
| Flotation with untreated BM | 50–57 | 79–95 | 58–69 | 17–46 |
| Fenton pretreatment (0.5 M) combined flotation | 48–74 | 33–65 | 34–51 | 48–82 |
| Roasting pretreatment combined flotation (optimum parameters) | 77 | 75 | 83 | 90 |

**Author Contributions:** H.Q. performed the experiments and was responsible for the analytical data preparation. C.P. and S.N. performed the SPME-GC-MS and pyrolysis-GC-MS measurements with the supervision of M.W., and J.K. was involved in the interpretation of the statistical experimental results and the evaluation of the flotation effect. H.Q. and S.N. prepared the manuscript. D.G. was responsible for the strategic approach and supervised the work. All authors helped to improve the manuscript. All authors have read and agreed to the published version of the manuscript.

**Funding:** This research was funded by the German Ministry on Education and Research (BMBF) within the research program: "ProZell 2" (InnoRec, reference number: 03XP0246A and 03XP0246C).

**Acknowledgments:** The authors acknowledge the financial support of the German Federal Ministry of Education and Research within the cluster project, "ProZell InnoRec". We thank Petra Sommer, Heike Grosse, and Maike Gamenik from the IFAD Institute of Mineral and Waste Processing, Waste Disposal and Geomechanics for the elemental analysis. We acknowledge the financial support by the Open Access Publishing Fund of the Clausthal University of Technology.

**Conflicts of Interest:** The authors declare no conflict of interest. No personal circumstances or interests that may be perceived as inappropriately influencing the representation or interpretation of the reported research results are given. The funders had no role in the design of the study; in the collection, analyses, or interpretation of data; in the writing of the manuscript; or in the decision to publish the results.

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
