# Peer review of "Recovery of Graphite and Cathode Active Materials from Spent Lithium-Ion Batteries by Applying Two Pretreatment Methods and Flotation Combined with a Rapid Analysis Technique"

_metals, doi:10.3390/met12040677_

Round 1

Reviewer 1 Report

Very nice piece of work.

Author Response

Overall response: We sincerely thank you for the comment!

Reviewer 2 Report

This manuscript titled Application of a Rapid Analysis Technique as tool for control and process optimization in chemical and thermal pre-Treatment and Flotation Separation of Graphite and NMC Particles from spent Li-ion Batteries, investigated the comprehensive recycling of two 15 products - graphite and NMC - via pre-treatment and flotation using specific analytical methods to 16 identify relevant influencing factors.

Although the paper has an original component, it is hardly distinguished in the abstract. Please, kindly re-write the abstract so that the innovative component of the study be emphasized.

Some suggestions and corrections in your work will make your manuscript more attractive and suitable to be published in Metals.

Firstly, some polishing is required in order to merit the publication level of this journal. Authors should revise the paper under the following amendments.

Please, make a shorter title, so the readership understand from the beginning the central idea of the research.

Please, pay attention in terms of English Grammar.

The authors should identify more clear and coherent the objective of the research. There are no very well-defined key points, as well.

Please, write a more concise state of the art, highlighting:

  1. What is the importance and scope of this work in the context of this journal? Need to discuss the relation/impact of the proposed work with the scope of Metals in introduction briefly.
  2. Abstract would be kept simple and precise with the novelty of the paper.

Reviewer 3 Report

This manuscript presents the results of investigations on a relevant subject matter of Journal «Metals». In manuscript methods for separating the components of lithium-ion batteries have been compared. The results are understandable, but should be improved for publication.

Comments

(I) Do not use an abbreviation in the title of an abstract article.

(II) The experiments were carried out in "small volumes" (125 mL). Why the units of measurement are tons (Table 1)?

(III) What is the accuracy of determining the content of metals in samples?

(IV) Figures 1 and 2: It is better to specify the particle size range of each fraction (0.5...1 mm or >2 mm).

(V) Figures 1 and 2: The original composition of the processed object should be indicated.

(VI) Figure 5: The parenthesis is missing (515 °C pyrolysis temperature).

(VII) On what basis were the concentrations selected for the Fenton process? Are they optimal?

(VIII) Are oxygen-containing organic compounds present in the samples after pre-treatments?

(IX) Conclusions: The authors did not conclude which pre-treatments of processed object are better.

In general, the topic of the manuscript is relevant and can be published after minor revision.

Reviewer 4 Report

Interesting study!

A few comments:

Throughout the manuscript inappropriate significant figures are presented. For example, in abstract a number like 89.51% indicates precision of measurement to ~1 part in 9,000.

L 134 - strength of H2O2?

Figs. 2 & 3: OK, I wanted to compare each size range in the two figures -- but they are plotted inversely. Best to get them so each size lines up in the 2 figures AND get the x-axis same length and bars same width. Maybe even put both figures on the same larger figure.

I don't think that the Conclusions did justice to the Title - Rapid Analysis Technique. Also, maybe a diagram or flowsheet of process control points and equipment in the Discussion would be useful to the reader. 

Is the recovery of graphite in the process(es) described here likely to be cost-effective? A few words on that might be of interest. 

Reviewer 5 Report

This manuscript is devoted to the investigation of two pretreatment methods for improving flotation separation of graphite and NMC particles from spent Li-ion batteries. The content of this paper is of topical importance and will be of interest to readers of metals. Thus, I'd like to recommend this paper to be considered for publication in metals after addressing the following comments.

  1. Section 2.3.3. Flotation experiment (p.5): Flotation experiments of samples pretreated by Fenton and roasting were conducted with different collector dosage. For the comparison of these two pretreatment techniques, flotation experiments should be conducted under the same conditions. Please explain why different collector dosages were used.
  2. Please inform the pH for flotation experiments.
  3. Section 3.3: At a higher concentration of Fenton's reagent (i.e., 0.5 M), the graphite recovery decreased. Please explain why it happened. Also, it is recommended to mention pH dependency of Fenton reaction.
  4. Lines 343-345: The authors mentioned that the SBR-based binders were not completed removed due to high S/L ratio. If so, why did the authors  not further investigate Fenton pretreatment with lower S/L ratio and/or higher concentration of Fenton's reagent?
  5. It is recommended to add a table summarizing the result of current work and others.

Round 2

Reviewer 5 Report

The authors revised the manuscript based on the reviewers' comments, but some of the answers are not satisfactory.

  • Points 3 and 4: These should be more clearly addressed. Discussions are very weak.
  • Point 6: I recommended the authors to compare the results of this work and other works, which will make the readers to grasp the significance of this study.
